# Consequences of COVID-19 Confinement for Teachers: Family-Work Interactions, Technostress, and Perceived Organizational Support

**DOI:** 10.3390/ijerph182111259

**Published:** 2021-10-27

**Authors:** Patricia Solís García, Rocío Lago Urbano, Sara Real Castelao

**Affiliations:** 1Faculty of Education, Universidad Internacional de La Rioja, 26006 Logroño, Spain; 2Psychology Department, Universidad de Huelva, 21004 Huelva, Spain; rocio.lago@dpee.uhu.es; 3Centro Asociado UNED Ponferrada, 24400 Ponferrada, Spain; sarreal@ponferrada.uned.es

**Keywords:** psychosocial factors, professional-private interplay, COVID-19 pandemic, well-being, teachers, education, technostress, perceived organizational support

## Abstract

The confinement experienced due to the COVID-19 pandemic has prompted a rethink of the teaching–learning process to which teachers have responded without planning and instead used their resources. This study aimed to analyze the relationships between work–family interactions, technostress, and perceived organizational support in teachers during the confinement period in Spain that began in March 2020. An online survey was administered to 640 pre-school, primary, and secondary school teachers. Positive reciprocal work–family interactions and their relationship with organizational support were found, with differences according to gender, with women showing a more negative perception of the impact on the family. There were no marked levels of technostress in the overall sample, although higher levels of perceived ineffectiveness and skepticism were found in teachers aged 46 years or older. Teachers in private and subsidized schools showed a higher level of perceived support than those in public schools. There is a need to continue this work to verify the values of these dimensions in other contexts and to apply institutional measures and public policies to improve these indicators in this group.

## 1. Introduction

Society is facing an unprecedented challenge caused by the social and health crisis resulting from the COVID-19 pandemic. As a fundamental part of society, educational institutions have undergone a change that has affected both teaching staff and the family members of those in school. The current social distancing strategies have entailed a complete reorganization of the educational system, with the period of confinement decreed by the COVID-19 health emergency being the moment requiring the greatest adaptation, which, in turn, has had significant consequences for teaching activities. According to UNESCO estimates (2020) [1], the number of children and young people around the world who have been affected by the closure of educational centers is at 1.2 billion. Thus, educational institutions have been forced to rapidly readapt their classes to a virtual format and undergo a complete digital transformation [2,3,4]. Teachers have had to resort to online platforms to reach students, webinars have become temporary classrooms, parents have monitored their children’s studies from home, and students have been deprived of peer-to-peer interaction [5].

During the confinement period, teachers have continued to play an essential role in directing students’ work [6]. However, this learning guidance was provided to students from their homes, and family factors therefore entered the picture, interacting very closely with the work dimension. The family dimension forms a vital pillar in the life of teachers and influences the welfare of all members [7]. Previous studies from the perspective of students found that family support during online teaching was the greatest predictor of academic resilience [8]. Likewise, teachers have been forced to implement ICT-based methodologies under the guidelines of their respective organizations; there are antecedents that show that perceived organizational support and self-efficacy have a significant positive influence on the readiness for change of teachers during the pandemic situation [9]. The adaptability of this group is also related to the experience in online teaching and perceived self-efficacy when using technologies [10]. For this reason, the following three pillars that have been fundamental in teaching during confinement are discussed below: teachers’ family interactions, the technostress associated with the use of technologies in teaching, and the level of perceived organizational support provided during the confinement period.

Before analyzing the results found in the surveyed sample, we describe each of the variables of interest for this study.

Research has been carried out on family–teacher interactions for more than three decades. This work has largely focused on the conflicts resulting from this interaction and its bidirectional nature [11]. Thus, it has been reported that work–family conflict, family satisfaction, and perceived family support are closely related [12]. Approaching this issue through the prism of the ecological systems theory, the support of coworkers and supervisors, as well as of the family environment itself, particularly the spouse, contributes to decreasing the negative work–family interaction, thus increasing positive interactions [13], acting as a significant protector against the impact of stressful life events on general health [14]. On the other hand, long working hours and overload prevent people from enjoying quality time with their families, which in turn results in a negative evaluation of their work [15,16]. In particular, teachers tend to consider that work detrimentally affects their family life [17]. Therefore, the priority in this interaction lies in the existence of a balance between the work environment and the family relationship [18].

Concerning learning in virtual environments, it is currently necessary to acquire digital competencies as part of the teaching–learning process to cope with the changes demanded by society. These competencies have been shown to be effective in responding to the needs of a globalized society [19]. During confinement, teachers have turned to the implementation of online learning, which has multiple benefits. These include higher academic attainment since such a learning environment provides an educational context in which academic management procedures are at the service of the student. In addition, the online environment for allows implementing collaborative in-network learning methodologies with available pedagogical and didactic resources; it gives value to tutorial actions in the development of cognitive and metacognitive competences, maintains procedural and permanent evaluation systems, and accommodates the accessibility, permanence, and progress of students, as well as promoting the use of technological and didactic tools that favor interactivity [20]. However, the use of ICT strategies in virtual training environments in compulsory school stages is still scarce and tends to depend on previous training and the availability of resources and access to the network among teachers and families [21]. In this sense, the use of ICT is not yet fully implemented and is considered an added barrier to the teaching process rather than a resource [22]. Moreover, even among those teachers who can use ICT in their work and personal life, there are still few who can be granted the status of innovator [23].

Concerning technostress, the rapid digitalization of society has generated a demand to accelerate the implementation of ICT in teaching, and changes in curriculum policies reflect this development [24]. While ICTs have benefits for teaching, there are negative factors linked to their use in the teaching work environment, such as the need—or obligation in some cases—to be constantly connected, which causes fatigue and stress, among other problems [25]. This type of stress has been termed technostress; a term that had already been proposed in 1984 by Brod [26] to refer to a modern disease that occurs when someone is perceived as unable to adapt to computer technologies. Technostress is currently considered a complex syndrome that poses an emerging risk [27]. In addition, while the research currently available on this topic is still limited [28,29], some studies warn that teachers, particularly at primary and infant levels, are suffering from high levels of general stress that affect their quality of life [30,31], a situation that might favor the development of specific stress resulting from the use of technologies—or, in other words, technostress.

For many teachers, employing ICT involves a substantial change in their usual way of engaging with and thinking about the teaching process, which generates stress [25,32]. Çoklar et al. [33] have pointed out five main causes of technostress in teachers: individual problems (self-efficacy, attitude, and economic situation), technical problems, education-oriented problems, health problems, and time problems. Syvänen et al. (2016) [24] found that key predictors of technostress are ICT competence, alignment of educational use of ICT with teaching style, school support, and attitudes toward educational use of ICT. In addition, technostress has psychosocial consequences not only at the individual but also at the organizational level, and these can be manifest in anxiety, job dissatisfaction, and depression [34]. At the organizational level, these outcomes translate into absenteeism and poorer job performance associated with the misuse of technology [35]. 

Encouraging teachers to incorporate technology into the teaching process fully requires training and adequate support from organizations and educational policies [32]. Next, we briefly explore the background on perceived organizational support.

Perceived organizational support refers to the belief and interpretation made by employees of an institution about how the organization itself values their contributions and cares about their well-being [36]. Thus, interpersonal relationships within a work organization are the result of organizational support; this entails exchanging information and contributing to the organizational structure to perceive themselves as valued [37]. If an employee perceives organizational support, their commitment to the institution will be increased, which will also increase the expectation of a reward based on this additional effort [38], which will serve as buffer against the stress perceived by the worker [39]. In contrast, organizational support has a negative relationship with burnout [40]. 

Given this background and the apparent interconnection between the issues described above, this study aimed to test the relationships between work–family interactions, technostress, and perceived organizational support in teachers during the confinement period imposed due to the COVID-19 social and health crisis.

## 2. Materials and Methods

The presented study was a quantitative, non-experimental, descriptive research design.

The sample consisted of 640 teachers (178 men and 462 women) from 35 provinces of Spain, grouped into 14 autonomous communities, the main ones being Castilla y León (25.5%), Castilla La Mancha (14.7%), Asturias (12.2%), Navarra (10.6%), Extremadura (10.3%), and Catalonia (7.5%). The schools are located in urban (60.5%) and rural (39.5%) areas.

Of the teachers surveyed, 75.9% worked in public schools, of which 63.58% indicated that they were career civil servants, and 35.39% were temporary (1.12 did not indicate the type of contract), 22.8% in subsidized schools (22.8%), and 1.3% in private schools. In the latter two cases 83.36% had a permanent contract, and 13.64% had a temporary contract or a contract for work/service.

The mean age of the sample was 43.54 (SD = 9.85) (43.19 for women and 44.44 for men), with a range of 23 to 64 years. The sample reported having a mean of 15.42 years (SD = 10.74) (15.10 for women and 16.25 for men) teaching experience. Concerning the training of the teachers in the sample, 33.6% had studied teaching in primary education (28.9% of them had completed a diploma, and 4.7% had completed a degree). In addition, 10% had studied teaching in Early Childhood Education (8.6% have a diploma and 1.4% have a degree). Of these, 21.1% had also completed the Pedagogical Accreditation Course, and 15.8% had completed the master’s degree in Teacher Training. Finally, 3.1% had also completed doctoral studies.

Regarding the educational stages in which they mainly taught, 41.2% taught in primary education, 28.6% in secondary education, 11.6% in early childhood education, 9.4% in training courses, and 9.2% in high school. Regarding the various functions they performed in the centers, 21.3% belonged to the management team, 28.4% were tutors, 21.7% were non-teaching primary school staff, 24.7% were non-teaching secondary school staff, and 3.9% held other positions.

The 640 teachers responded to the items of 3 instruments that evaluated the variables described above.

Three types of instruments were applied in the study.

Survey Work-Home Interaction-Nijmegen (SWING): Created by Geurts (2001) [41], this assesses the relationship between work and family. It consists of 22 items distributed across four subscales: negative work–family interaction, negative family–work interaction, positive work–family interaction, and positive family–work interaction. The instrument has been validated for the Spanish population [42], demonstrating good psychometric properties and a good internal consistency, with values between 0.77 and 0.89. It has been employed in previous research with teachers [17,43].

Eisenberger and his team developed the Survey of Perceived Organizational Support (SPOS) in 1986 [39] and assessed workers’ beliefs regarding the perceived level of support they received from the organization where they worked. Its abbreviated version consists of 17 items with a Likert-type scale of 7 response options, from “strongly disagree” to “strongly agree”; it is unidimensional and has a Cronbach’s Alpha of 0.93. This version has been adapted and validated in Spanish [44], showing a Cronbach’s Alpha of 0.78. The instrument was subsequently used by others [45,46].

The RED-TIC Technostress Questionnaire [47]. This instrument includes four blocks of variables (administered data, ICT use, psychosocial risks, and psychosocial consequences). It consists of 4 dimensions: fatigue, anxiety, skepticism, and ineffectiveness. It also presents adequate internal consistency with Alpha values above 0.83 in all dimensions and has been used by others [15,48,49].

The management teams of educational centers throughout Spain were contacted and provided with information on the content and objectives of the study. Furthermore, a digital link was made available during April, May, and June 2020, so that interested teachers could participate by completing the questionnaire anonymously and voluntarily without reward, thus complying with the requirements of the Research Ethics Committee. Furthermore, to avoid potential bias in the responses, the participants were assured that there would be no individual analysis of their responses.

Before carrying out the subsequent analyses, the assumptions of normality and equality of variances (homoscedasticity) were tested using the Kolmogorov–Smirnov and Shapiro–Wilk statistics, Q–Q plots, and the Levene test (based on means). Since the sample did not meet the assumptions of normality, non-parametric tests were used.

## 3. Results

First, the total scores were calculated for the variables negative work–family interaction, negative family–work interaction, positive work–family interaction, positive family–work interaction, positive family–work interaction, perceived organizational support, inefficacy, skepticism, fatigue, and anxiety (these last four being linked to technostress). Thus, it was found that, broadly speaking, the sample perceived positive reciprocal work–family (M = 1.55, SD = 0.63, range 0–3) and family–work (M = 2.17, SD = 0.61, range 0–3) interactions. Perceived organizational support reached an overall mean score of 4.20 (SD = 0.51, range 1–7). The levels of technostress were distributed as follows: ineffectiveness (M = 1.43, SD = 1.33, range 0–6), skepticism (M = 1.69, SD = 1.63, range 0–6), fatigue (M = 1.29, SD = 0.86, range 0–6), and anxiety (M = 0.51, SD = 0.53, range 0–6).

### 3.1. Differences According to Sociodemographic Variables

Next, we explored the existence of differences in terms of gender (male and female), coded age (up to 45 years and 46 or more years), and type of center (public, subsidized-private) in terms of work–family interactions, perceived organizational support, and technostress, using the Mann–Whitney U test for these categorical variables.

Concerning gender, no significant differences were found, except for the negative work–family interaction variable, in which women perceived this interaction more strongly, with the belief that their current teaching work had a negative impact on their family life.

Concerning the age variable, no significant differences were found. However, it should be noted that younger teachers (up to 45 years of age) showed higher mean scores for all variables except perceived ineffectiveness and skepticism, both of which are linked to technostress. Teachers aged 46 or older perceived that they had fewer skills related to new technologies and were doubtful about the value of such skills.

For the analysis according to the type of center, this variable was re-coded by establishing two categories (public center and private-subsidized center) (Table 1). Significant differences were found in the following variables concerning the negative family–work interaction. First, teachers in public schools showed a greater perception that their family life was a burden for the development of their work. Second, concerning perceived organizational support, teachers in private-subsidized schools perceived greater support from the school’s management and felt that they were more highly valued by the latter. Finally, teachers in public schools perceived more anxiety related to technostress when teaching during confinement.

### 3.2. Correlations between Different Questionnaires

Non-parametric correlations were conducted using Spearman’s rho test (Table 2). Perceived organizational support was found to correlate significantly at the 0.05% level with perceived fatigue linked to technostress and perceived organizational support (r = −0.81). A correlation was also found between negative family–work interaction (r= −0.92) and positive family–work (r = 0.18) and work–family (r = −0.22) interactions and perceived organizational support.

On the other hand, regarding positive interactions, positive work–family interaction correlated with fatigue linked to technostress (r = −0.19) and positive work–family interaction correlated with perceived inefficacy (r = −0.11). Concerning negative interactions, the negative work–family interaction correlated with inefficacy (r = 0.12), skepticism (r = 0.09), fatigue (r = 0.96), and anxiety (r = 0.47). On the other hand, the negative family–work interaction correlated significantly with the variables of inefficacy (r = 0.20), skepticism (r = 0.14), fatigue (r = 0.47), and anxiety (r = 1).

## 4. Discussion

The implications of the initial confinement resulting from the COVID-19 pandemic have been unpredictable and diverse. In the educational system, the necessary suspension of face-to-face teaching activities that have given way to virtual environments has placed high pressure on educational agents to adapt to this new scenario [6]. This drastically new approach of redesigning the teaching and learning process requires attending to the needs of all participants [50]. Thus, the tools and contents supported and presented as optional in many classrooms and levels became the main and only means of learning and communication between teachers and families. This unplanned situation also meant that institutions were unable to provide the necessary resources to support teachers and students (and their families) in these tasks.

Therefore, this study aimed to explore the relationships between work–family interactions, technostress, and perceived organizational support in teachers during the confinement initiated in Spain in March 2020.

First, concerning positive reciprocal work–family interactions and their relationship with perceived organizational support, it is important to highlight that, in the present study, women perceived that the development of their teaching work during confinement harmed their family life. Indeed, previous studies have also found differences according to gender [13]. When people achieve a work–family balance, their evaluation of their work is more positive, perceiving that their organization enables them to respond to both their work and family demands [25,51]. Therefore, organizational policies facilitating such reconciliation are essential [17]. 

Second, teacher technostress is important because of the intensity of ICT integration in teaching [10,11,12,13,14,15,16,17,18,19,20,21,22,23,24]. In this study, it was found that, although the levels of technostress were not pronounced, contrary to other studies that found technological overload, daily work intensity, techno-invasion, and socio-emotional consequences [52], the highest levels were recorded for the variables of ineffectiveness and skepticism, these scores being even higher for teachers aged 46 years or older. This makes it necessary to develop and implement an intervention program focused on stress coping strategies to help decrease technostress and enhance teachers’ teaching efficacy [23,53].

Finally, the relationship between perceived organizational support and its implications for commitment has been verified in previous works [54,55]. In the present study, the scores on this variable were shown to reach moderate levels, in line with the results of other studies [9,10,11,12,13,14,15,16,17,18,19,20,21,22,23,24,25,26,27,28,29,30,31,32,33,34,35,36,37,38,39,40,41,42,43,44,45,46,47,48,49,50,51,52,53,54,55,56]. Interestingly, perceived organizational support was lower in public school teachers, who also perceived more anxiety linked to technostress when teaching during confinement and greater negative family–work interactions.

## 5. Conclusions

The impossibility of foreseeing a change such as that required by the Covid-19 confinement has made it necessary to evaluate the impact of key variables on teacher performance, such as family–work interactions, technostress, and perceived organizational support. 

However, the scenario experienced has generated a learning experience that allows us to think about strategies and resources for improving the quality of teaching in situations in which teachers may have a greater need for reconciliation between family and schoolwork, use of digital media, and improvement of institutional relations within the workplace so that there is greater perceived organizational support. Taking these variables as dimensions, even if the confinement situation is not repeated (although this cannot be ruled out), this experience has prompted us to reflect on measures that could be taken to address the results reported here. 

As shown in the literature, measures to support work–life balance, digital competencies, and the improvement of the work environment could serve as a buffer against stress in general and the stress generated by using technologies in particular. Therefore, within the Spanish context, there is a need to consider actions that could support teachers who have performed their work without planning, relying on their own resources and, in many cases, who are individuals required to support the education of their own children. There is a particular need for strategies that could help teachers to provide quality education without suffering a decrease in their own quality of life—a condition that makes it impossible to perform their task with the dedication that a teaching–learning process requires, either in a widespread situation such as a global pandemic or in other unique situations that a teacher must face during their career. 

Research in this area should continue to evaluate the effects of the pandemic and to confirm whether, in similar situations (general or particular), there is now a better response from teachers because of their experience and resilience. Nonetheless, and as we have noted, personal experience will not be enough without the support of institutions and the implementation of public social and educational policies aimed at improving the well-being of teaching staff.

## Figures and Tables

**Table 1 ijerph-18-11259-t001:** Contrast statistics: Group differences in variables according to type of center.

Variables	Mann–Whitney U	W for Wilcoxon	Z	Sig. Asymptotic. (Bilateral)
Negative Work–Family interaction	34,549.500	46,484.500	−1.438	0.150
Negative Family–Work interaction	31,701.000	43,636.000	−0.926	0.003
Positive Work–Family interaction	34,284.000	152,625.000	−1.576	0.115
Positive Family–Work interaction	35,615.500	153,956.500	−0.908	0.364
Total SPOS	27,916.500	146,257.500	−4.758	0.000
Technostress Ineffectiveness	36,627.000	154,968.000	−0.400	0.689
Technostress Skepticism	35,212.500	47,147.500	−1.117	0.264
Technostress Fatigue	35,006.000	46,941.000	−1.213	0.225
Technostress Anxiety	31,701.000	43,636.000	−2.926	0.003

**Table 2 ijerph-18-11259-t002:** Spearman’s rho correlations.

Dimensions		Positive Work–Family Interaction	Positive Family–Work Interaction	Technostress Ineffectiveness	Technostress Skepticism	Technostress Fatigue	Technostress Anxiety
Positive Interaction Work–Family Interaction	C. of correlation	1.000	0.511 **	−0.066	−0.029	−0.196 **	−0.002
	Sig. (bilateral)	-	0.000	0.097	0.463	0.000	0.957
Positive Family–Work Interaction	C. of correlation	0.511 **	1.000	−0.110 **	−0.030	−0.041	−0.069
	Sig. (bilateral)	0.000	-	0.005	0.446	0.296	0.083
Technostress Ineffectiveness	C. of correlation	−0.066	−0.110 **	1.000	0.474 **	0.117 **	0.209 **
	Sig. (bilateral)	0.097	0.005	-	0.000	0.003	0.000
Technostress Skepticism	C. of correlation	−0.029	−0.030	0.474 **	1.000	0.079 *	0.149 **
	Sig. (bilateral)	0.463	0.446	0.000	-	0.044	0.000
Technostress Fatigue	C. of correlation	−0.196 **	−0.041	0.117 **	0.079 *	1.000	0.476 **
	Sig. (bilateral)	0.000	0.296	0.003	0.044	-	0.000
Technostress Anxiety	C. of correlation	−0.002	−0.069	0.209 **	0.149 **	0.476 **	1.000
	Sig. (bilateral)	0.957	0.083	0.000	0.000	0.000	-

**. The correlation is significant at the 0.01 level (bilateral). *. The correlation is significant at the 0.05 level (bilateral).

## Data Availability

The data that support the findings of this study are available from the corresponding author upon reasonable request.

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
