# Peer review of "Consequences of COVID-19 Confinement for Teachers: Family-Work Interactions, Technostress, and Perceived Organizational Support"

_ijerph, 2021, doi:10.3390/ijerph182111259_

Round 1

Reviewer 1 Report

Very relevant article given the contingency product of the Covid-19 pandemic, therefore it contributes to scientific knowledge.

Regarding the structure, the results mention and describe the statistical analysis techniques, however, this information must be presented in the materials and methods.

It is recommended to clarify the type of research and strengthen the discussion.

Author Response

Following the suggestions of reviewer number 2, a sentence has been introduced indicating the type of investigation and the statistical analysis techniques are described in the materials and methods section. Likewise, the discussion has been strengthened with new studies.

Reviewer 2 Report

A nice and interesting topic, compliments for the authors.

I have one minor remarks:

Introdution, second paragraph on page 2:

I consider the rational for the study als rather weak, why are these pillars fundamental? I suggest the authors to motivate the reasoning more strongly, supported by some more literature.

Author Response

the introduction has been reinforced, supporting in the literature the convenience of the 3 fundamental pillars of the study.